# Regeneration in Mice of Injured Skin, Heart, and Spinal Cord by α-Gal Nanoparticles Recapitulates Regeneration in Amphibians

**DOI:** 10.3390/nano14080730

**Published:** 2024-04-22

**Authors:** Uri Galili, Jianming Li, Gary L. Schaer

**Affiliations:** Department of Medicine, Rush University Medical Center, Chicago, IL 60612, USA; jli@medtricbiotech.com (J.L.); gary_schaer@rush.edu (G.L.S.)

**Keywords:** α-gal nanoparticles, scar-free regeneration, anti-Gal, skin regeneration, myocardial regeneration, spinal cord regeneration, complement activation

## Abstract

The healing of skin wounds, myocardial, and spinal cord injuries in salamander, newt, and axolotl amphibians, and in mouse neonates, results in scar-free regeneration, whereas injuries in adult mice heal by fibrosis and scar formation. Although both types of healing are mediated by macrophages, regeneration in these amphibians and in mouse neonates also involves innate activation of the complement system. These differences suggest that localized complement activation in adult mouse injuries might induce regeneration instead of the default fibrosis and scar formation. Localized complement activation is feasible by antigen/antibody interaction between biodegradable nanoparticles presenting α-gal epitopes (α-gal nanoparticles) and the natural anti-Gal antibody which is abundant in humans. Administration of α-gal nanoparticles into injuries of anti-Gal-producing adult mice results in localized complement activation which induces rapid and extensive macrophage recruitment. These macrophages bind anti-Gal-coated α-gal nanoparticles and polarize into M2 pro-regenerative macrophages that orchestrate accelerated scar-free regeneration of skin wounds and regeneration of myocardium injured by myocardial infarction (MI). Furthermore, injection of α-gal nanoparticles into spinal cord injuries of anti-Gal-producing adult mice induces recruitment of M2 macrophages, that mediate extensive angiogenesis and axonal sprouting, which reconnects between proximal and distal severed axons. Thus, α-gal nanoparticle treatment in adult mice mimics physiologic regeneration in amphibians. These studies further suggest that α-gal nanoparticles may be of significance in the treatment of human injuries.

## 1. Introduction

The objective of this review is to describe studies performed in the last 14 years which support the hypothesis that α-gal nanoparticles applied to external and internal injuries in adult mice can induce immune-mediated regenerative processes which are naturally occurring in urodeles. The amphibian urodeles, including salamander, newt, and axolotl, display the unique ability among vertebrates of regenerating an amputated limb [1,2,3]. One of the early events following the amputation of urodele limbs is the recruitment of macrophages into the amputation area. These recruited macrophages induce further migration of fibroblasts and other cells into the stump, proliferation, and dedifferentiation of cells into progenitor cells, forming the blastema tissue in which a variety of progenitor cells differentiate into tissues that rebuilt the amputated limb [1,2,3,4,5,6,7]. These regenerative processes are complex, with multiple cell–cell, cytokine–cell, and extracellular matrix (ECM)–cell interactions [4,5,6]. Despite this complexity, it is well established that without the initial recruitment of macrophages to the amputation site, no further limb regeneration takes place [1,4,5,6,7]. Similar macrophage-mediated processes have been observed in scar-free regeneration in the injured heart, skin, and spinal cord of urodeles [8,9,10,11]. Mammals lack the ability to regenerate amputated limbs as well as most of their injured tissues, including skin wounds, damaged ventricular walls post-myocardial infarction, and injured central and peripheral nerve systems. Although the healing of wounds and myocardial injuries in adult mammals also involves early migration macrophages into the damaged tissue, the healing process results in fibrosis and scar formation rather than in the restoration of the normal structure and function of the injured tissues [12,13,14,15,16,17]. Thus, based on the results of their activity, macrophages mediating regeneration are referred to here as “pro-regenerative” macrophages [18], whereas macrophages mediating healing and repair by default fibrosis and scar formation are referred to as “pro-reparative” macrophages.

The macrophage-mediated regenerative mechanisms observed in urodeles may have been partially conserved in mammals, as suggested by regenerative processes in some mammalian fetuses and neonates. Studies on wound healing in the skin of mouse fetuses have demonstrated scar-free regeneration which is associated with extensive macrophage migration into the wounds [19,20,21]. Moreover, resection of the heart apex in mouse neonates (i.e., on the first and second days after birth) was followed by scar-free regeneration and restoration of normal contractile function of the injured myocardium within 2–3 weeks [22,23,24]. A similar regenerative capacity was observed in the injured myocardium of porcine neonates [25,26]. As in limb and heart regeneration in urodeles, the regeneration of injured myocardium in mouse neonates was characterized by initial extensive migration of macrophages into the injury site in the heart, followed by proliferation of cells that matured into cardiomyocytes [24]. In mice older than 7 days, repair of the injured myocardial apex was also found to be associated with extensive infiltration macrophages, but in contrast to neonatal mice, the infiltrating macrophages induced repair by fibrosis and scar formation [22,23,24]. Similar macrophage-induced fibrosis and scar formation was observed in adult mice [14,15,16] and in humans following myocardial infarction (MI) [17]. These observations have led to the assumption that the regeneration-inducing capacity of macrophages, observed in urodeles, has been evolutionarily conserved in the early stages of mammalian life, but it is suppressed within few days after birth. Thus, it was suggested that identifying the causes for this suppression may restore the regenerative potential in the healing of injuries in adult mice [27,28].

One of the common characteristics in the early steps of injured tissue regeneration in urodeles and mouse neonates was found to be innate activation of the complement system in the absence of antigen/antibody interactions [29,30,31]. The complement system, which is a primordial sentinel of the innate immune response, becomes activated during tissue injury and remodeling, in particular, in lower vertebrates such as fish and amphibians. It was conserved even in the fetal and neonatal stages of mammalian evolution [31] and to a much lesser extent in very few tissues in adults, such as in injured liver and bone [30]. An analysis of genes activated in individual cells during blastema formation in the axolotl, following wound epidermis formation and macrophage infiltration, revealed a dynamic process in which a wide range of activated genes characteristic of progenitors of epidermal, mesenchymal, and hematopoietic lineages could be detected at various stages of differentiation [4,32]. Complex cell–cell and cell–intercellular matrix interactions ultimately result in the differentiation of the recruited cells into the regenerated limb. The molecular mechanism that triggers the complement cascade activation in injured urodeles, in the absence of any antigen/antibody interaction, is not clear yet but is thought to be the result of the release of unknown substances from damaged tissues [30]. The innate complement activation in mouse neonates vs. its absence in most adult mouse tissues suggests that the gene(s) associated with triggering this complement activation are suppressed in many tissues of adult mice.

A highly effective physiologic mechanism for localized activation of the complement system in adult mice and other mammals is the antigen/antibody interaction. Therefore, we hypothesized that extensive localized activation of the complement system by antigen/antibody interactions within injuries of adult mice may reactivate the suppressed ability of macrophages to become “pro-regenerative” cells which induce regenerative processes such as those observed in urodeles and neonatal mice. This assumption was studied by performing in situ interaction between the anti-Gal antibody and α-gal nanoparticles presenting a carbohydrate antigen called the “α-gal epitope” [28,33,34,35].

Anti-Gal is an abundant natural antibody in humans, constituting ~1% of circulating immunoglobulins [36,37,38,39]. It is produced throughout life in response to continuous antigenic stimulation by gastrointestinal bacteria [40,41,42]. Anti-Gal specifically binds α-gal epitopes with the structure Galα1-3Galβ1-4GlcNAc-R [43,44,45]. The α-gal epitope is abundantly synthesized by the glycosylation enzyme α1,3galactosyltransferase (α1,3GT) in non-primate mammals, lemurs, and New World monkeys [46,47]. In contrast, Old World monkeys, apes, and humans lack the α-gal epitope, but produce the natural anti-Gal antibody [46,47]. The reason for this distribution in mammals is an evolutionary selection process for the survival of ancestral Old World monkeys and apes which lacked α-gal epitopes and produced the natural anti-Gal antibody [48].

α-Gal nanoparticles (previously called α-gal liposomes [33,34]) are biodegradable nanoparticles presenting multiple α-gal epitopes. The binding of anti-Gal to the α-gal epitopes on these nanoparticles results in a localized extensive activation of the complement system [33,34,49]. The α-gal nanoparticles are comprised of phospholipids, cholesterol, and glycolipids, most of which present α-gal epitopes, which were all extracted from rabbit red blood cell (RBC) membranes in a chloroform:methanol solution [33,34,35]. After drying of the phospholipid, cholesterol, and glycolipid extract, the mixture was resuspended in saline by extensive sonication, resulting in the formation of submicroscopic liposomes (i.e., nanoparticles) presenting multiple α-gal epitopes (Figure 1A). The nanoparticles have an average size of ~300 nm, an average zeta potential of ~27 mV, and a polydispersity index of 0.988 [50].

The suggested regenerative effects of anti-Gal/α-gal nanoparticle interactions were studied in adult α1,3galactosyltransferase knockout mice (GT-KO mice) lacking α-gal epitopes and producing the anti-Gal antibody [51,52,53,54]. The α1,3GT gene *GGTA1* was inactivated in these mice by disruption (i.e., knockout) [51,55]. The GT-KO mice lived in a sterile environment and ate sterile food, and thus, they could not establish the gastrointestinal bacterial flora that stimulates the immune system for production of the natural anti-Gal antibody in humans [40]. Production of anti-Gal in titers comparable to those in humans is feasible by immunizing GT-KO mice with xenograft cells or cell membranes rich with α-gal epitopes like rabbit RBCs [56], porcine lymphocytes [53], or pig kidney membranes [33,34,35,52,54]. The hypothesis described below was studied in anti-Gal-producing adult GT-KO mice with skin, heart, and spinal cord injuries.

**Figure 1 nanomaterials-14-00730-f001:**
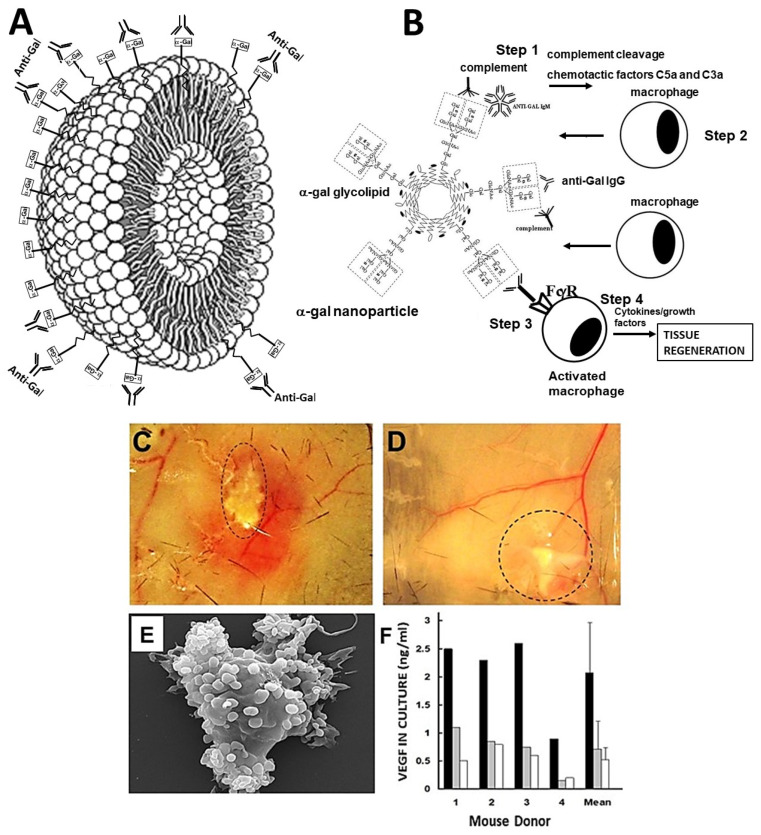
α-Gal nanoparticle structure and activity. (**A**) Illustration of an α-gal nanoparticle and the natural anti-Gal antibody it binds. α-Gal nanoparticles are submicroscopic liposomes comprised of a phospholipid bilayer “studded” with glycolipids presenting α-gal epitopes (rectangles) and cholesterol (not shown). (**B**) Hypothesis: α-Gal nanoparticles administered into injured tissues bind anti-Gal and activate the complement system (Step 1). Generated complement cleavage chemotactic peptides C5a and C3a recruit macrophages to the injury site (Step 2). The Fc portion (“tail”) of anti-Gal-coating α-gal nanoparticles binds to Fcγ receptors (FcγR) of recruited macrophages, activating them to polarize into pro-regenerative macrophages (Step 3). The activated macrophages orchestrate the regeneration of the injured tissue by secreting pro-regenerative cytokines/growth factors (Step 4). (**C**) Localized dilation of blood vessels by the activated complement system, 48 h after intradermal injection of 10 mg α-gal nanoparticles, as viewed on the basal surface of the skin. (**D**) Injection of 10 mg of control nanoparticles lacking α-gal epitopes results in no vasodilation. Dashed lines indicate the border area of the injected nanoparticles. (**E**) Scanning electron microscopy (SEM) of a macrophage binding of the α-gal nanoparticles (the small spheres attached to the macrophage), following 2 h of co-incubation. (**F**) Macrophages binging anti-Gal-coated α-gal nanoparticles secrete vascular endothelial growth factor (VEGF). Macrophages cultured with anti-Gal-coated α-gal nanoparticles (closed columns), macrophages and α-gal nanoparticles without anti-Gal (gray columns), or macrophages alone (open columns). Data from 4 GT-KO mice and means +SDs. Adapted with permission from Ref. [57]. 2018, Elsevier.

## 2. Hypothesis on the Regeneration of Injured Tissues in Adult Mice by α-Gal Nanoparticles

We hypothesized that the natural macrophage-mediated regeneration observed in injured tissues of urodeles and of mouse neonates can be reactivated in adult mice by the treatment of the injuries with α-gal nanoparticles [28,33,34,35]. This hypothesis is illustrated in Figure 1B in which the α-gal nanoparticle-induced regenerative process consists of several steps. In Step 1, α-gal nanoparticles administered into injuries of GT-KO mice producing the anti-Gal antibody bind this antibody via the multiple α-gal epitopes on the nanoparticles. Similar to most antigen/antibody interactions, anti-Gal binding to α-gal nanoparticles results in robust localized activation of the complement system. In the course of cascade activation of the complement system, cleavage of C5 and C3 complement components results in formation of C5a and C5b and of C3a and C3b. C5a and C3a complement cleavage peptides are highly potent chemotactic factors which direct the migration of macrophages. In Step 2, the newly generated C5a and C3a peptides direct, by their gradient of increasing concentration, the rapid recruitment of monocyte-derived macrophages to the injury site. In Step 3, the recruited macrophages reaching α-gal nanoparticles within the injury bind with high affinity, via their Fcγ-receptors (FcγR), the Fc “tails” of anti-Gal coating the α-gal nanoparticles. This binding induces the macrophages to polarize into pro-regenerative macrophages. In Step 4, the pro-regenerative macrophages are activated to secrete a wide range of cytokines/growth factors that orchestrate regeneration of structure and function of the injured tissue, similar to that observed in urodeles. This regenerative process also prevents occurrence of the default healing process of fibrosis and scar formation at the injury site. As described below, this hypothesized regenerative process could be demonstrated in adult anti-Gal-producing mice that had skin wounds, injured heart muscle due to myocardial infarction, and spinal cord injured by crushing.

The hypothesis is summarized as the following:Activation of the complement system occurs by anti-Gal binding to α-gal nanoparticles administered to the injury site.Macrophages are recruited to the injury site by the chemotactic complement cleavage peptides C5a and C3a.Binding of anti-Gal-coated α-gal nanoparticles to the recruited macrophages activates them to polarize into pro-regenerative macrophages.The pro-regenerative macrophages produce cytokines/growth factors that orchestrate regeneration of the treated injured tissue.

## 3. Experimental Demonstration of the Various Steps in the Hypothesis

*Activation of the complement system—*Intradermal injection of 10 mg α-gal nanoparticles in anti-Gal-producing GT-KO mice is immediately followed by anti-Gal binding to these nanoparticles. This interaction induces extensive localized activation of the complement system, as described in Step 1 in Figure 1B. This activation results in complement-mediated vasodilation around the α-gal nanoparticle-injected area, which was clearly observed 48 h post-injection (Figure 1C) [57]. Nanoparticles that lack α-gal epitopes (produced from the RBC membranes of GT-KO pigs) do not bind anti-Gal; therefore, the complement system is not activated and does not induce vasodilation in the vicinity of the nanoparticles lacking α-gal epitopes (Figure 1D).

*Recruitment of macrophages—*The complement cleavage peptides C5a and C3a generated as a byproduct of the complement cascade activation in Step 1 induce rapid recruitment of macrophages to the nanoparticles (Step 2 in Figure 1B). This recruitment is observed within 24 h post-injection (Figure 2A) and is significantly faster than that observed in untreated injuries in mice [34] in which macrophage migration is directed by MCP1, MIP1, and RANTES [58,59,60,61,62,63,64,65]. The α-gal nanoparticles are absent at the injection site because they are solubilized by ethanol during fixation. The number of recruited cells increased after 4 days and they were all stained by the macrophage-specific F4/80 antibody [34]. The recruited macrophage number peaked by Day 7 (Figure 2B) and remained stable after 14 days. However, all macrophages disappeared by Day 21, and the injection areas displayed normal structures of the skin with no indication of granuloma or inflammatory reaction [34]. Recruitment of macrophages by α-gal nanoparticles has been observed in additional sites of injection, including near a branch of the sciatic nerve (Figure 2C), spinal cord [66], and myocardium [35]. Inhibition of complement activation by cobra venom factor administered together with the α-gal nanoparticles resulted in no macrophage recruitment to injection sites [34].

*Characterization of the recruited macrophages*—Macrophages recruited into biologically inert polyvinyl alcohol (PVA) sponge discs, containing 10 mg α-gal nanoparticles, were harvested 7 days post-subcutaneous implantation, and were subjected to further characterization. Each of the sponge discs contained ~0.5 × 10^6^ cells, most of which displayed the morphology of large cells with ample cytoplasm filled with vacuoles that contained internalized anti-Gal-coated α-gal nanoparticles (Figure 2D) [34]. These nanoparticles cannot be observed because they are solubilized by ethanol during fixation. Flow cytometry analysis of these cells confirmed that the large majority are macrophages as they expressed the macrophage-specific markers CD14 and CD11b (Figure 2E) [33]. In contrast, no T or B lymphocytes were found to be recruited by α-gal nanoparticles. Most of these recruited macrophages were found to be at the M2 polarization state, as they were IL-10^pos^, arginase-1^pos^, and IL-12^neg^ (Figure 2F) [67]. Control sponge discs with only saline contained ~0.2 × 10^5^ cells, which were <5% of the number of macrophages recruited by α-gal nanoparticles [33].

Five-day culture of the macrophages recruited by α-gal nanoparticles resulted in the formation of cell colonies at a frequency of one colony per 50,000 to 100,000 cultured macrophages (Figure 2G,H) [57]. Each of the colonies included 300–1000 cells. The numbers of cells in these colonies implied that the cells forming them proliferated at an average cell cycle time of ~12 h. Detachment of the cells in these colonies and their analysis by flow cytometry following immunostaining indicated that they presented the mesenchymal stem cell (MSC) markers Sca-1 and CD-29 (Figure 2I,J) [57,68]. These findings strongly suggest that the M2 macrophages recruited and activated by α-gal nanoparticles direct the recruitment of MSCs to the site of the administered nanoparticles.

**Figure 2 nanomaterials-14-00730-f002:**
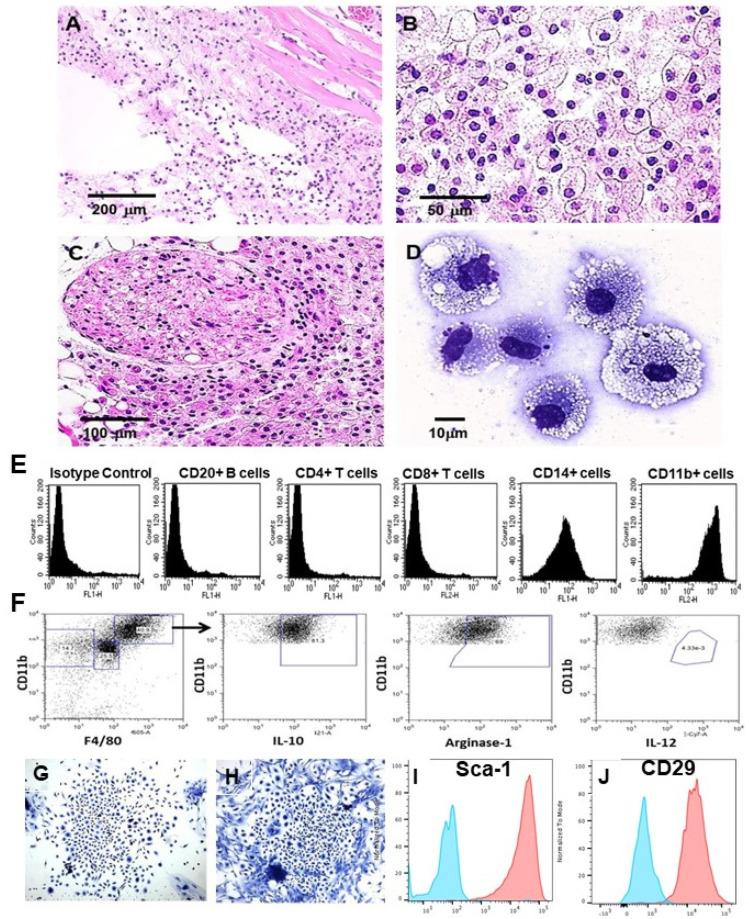
Macrophage recruitment and activation following administration of 10 mg α-gal nanoparticles. (**A**) Multiple macrophages migrate to the injection site within 24 h post-injection of α-gal nanoparticles (H&E ×100). (**B**) Macrophage number increases at the injection site after 7 days. The activated macrophages display a large size and ample cytoplasm (H&E ×400). (**C**) Macrophages recruited to a branch of the sciatic nerve area 4 days post-administration of 10 mg α-gal nanoparticles close to the nerve (H&E ×200). (**D**) Macrophages harvested 7 days post-implantation from PVA sponge discs containing 10 mg α-gal nanoparticles and implanted subcutaneously (×1000). (**E**) Flow cytometry analysis of cells recruited by α-gal nanoparticles into PVA sponge discs, 7 days post-implantation. Most infiltrating cells are macrophages characterized by the expression of CD11b and CD14, whereas no significant infiltration of T or B cells is observed (representative from 5 mice). (**F**) Flow cytometry analysis of the polarization state of large macrophages recruited by α-gal nanoparticles into PVA sponge discs, as in (**E**). Most of the macrophages are M2-polarized, as indicated by positive staining for arginase-1 and for IL-10 and negative staining for IL-12. (**G**,**H**) Cell colonies growing for 5 daysof culturingof fromcells recruited by PVA sponge discs containing α-gal nanoparticles, and presented in (**D**) (×100). (**I**,**J**) Expression of MSC markers Sca-1 and CD-29, respectively, by cells harvested from colonies like those in (**G**,**H**) and presented as orange curves. Isotype controls are presented as blue curves. Adapted with permission from Ref. [57] 2018, Elsevier, and [68] 2023, MDPI.

*Binding of α-gal nanoparticles to recruited macrophages*—Once the recruited macrophages reach the anti-Gal-coated α-gal nanoparticles, the Fc “tail” of this antibody can bind to Fc receptors (FcγR) on macrophages (Step 3 in Figure 1B). This is demonstrated in Figure 1E in which macrophages were incubated in vitro with anti-Gal-coated α-gal nanoparticles for 2 h. The nanoparticles bound the macrophages via Fc/FcγR interaction and covered the surfaces of the macrophages [34,49]. α-Gal nanoparticles lacking bound anti-Gal are not capable of binding to macrophages (not shown) [57].

*Activation of macrophages by bound α-gal nanoparticles—*The binding of anti-Gal-coated α-gal nanoparticles to macrophages induces their activation (Step 4 in Figure 1B). One of the cytokines tested to be produced as a result of this activation is VEGF. Whereas macrophages were incubated for 24 h at 37 °C with α-gal nanoparticles, those lacking anti-Gal displayed only background levels of VEGF production, while macrophages co-incubated with anti-Gal-coated α-gal nanoparticles produced VEGF at levels that were 2.5–4 fold higher than the background levels (Figure 1F) [34]. Studies on qRT-PCR with macrophages recruited in the skin by intradermal injection of α-gal nanoparticles (as in Figure 2A,B) further indicated that these cells displayed activation of fibroblast growth factor (FGF), interleukin 1 (IL1), platelet-derived growth factor (PDGF), and colony-stimulating factor (CSF) [34]. These findings suggest that the binding of anti-Gal-coated α-gal nanoparticles to macrophages induces the production of a range of cytokines/growth factors that can mediate the regeneration of injured tissues.

## 4. Accelerated Scar-Free Regeneration of α-Gal Nanoparticle-Treated Wounds

In order to determine whether α-gal nanoparticles alter physiologic wound healing, full-thickness dorsal excisional oval wounds (~9 × 6 mm) were created in anesthetized anti-Gal-producing GT-KO mice. The wounds were covered with a spot adhesive wound dressing that adsorbed 0.1 mL saline containing 10 mg α-gal nanoparticles, whereas control wounds were covered with a dressing that adsorbed only saline. Wounds covered with a dressing coated with α-gal nanoparticles displayed 95–100% healing (i.e., covered with regenerating epidermis) within 6 days (Figure 3A,B,D). Control wounds displayed only residual healing after 6 days (Figure 3A–C) and complete healing was observed only after 14–15 days (Figure 3B) [34,49]. An additional control was that of wounds treated with nanoparticles lacking α-gal epitopes, produced from RBCs of GT-KO pigs. These nanoparticles had no accelerating effect on wound healing, which was similar to that in wounds of mice treated with saline (Figure 3A).

Healing of wounds treated with α-gal nanoparticles was also evaluated in anti-Gal-producing GT-KO mice by the dorsal excisional splinted wound model. In this model, a splinting ring tightly adheres to the skin around the wound, preventing wound closure caused by skin contraction [69]. This model allows wounds to heal in a process similar to that in humans (i.e., without contraction of the wound). Also in these studies, α-gal nanoparticles induced accelerated wound healing and closure associated with accelerated rates of keratinization, vascular growth, and wound tissue deposition [70]. A similar acceleration of full-thickness wound healing by α-gal nanoparticles was observed in GT-KO pigs that lacked α-gal epitopes and produced the natural anti-Gal antibody [71].

The wounds treated with α-gal nanoparticles or saline were further examined histologically 4 weeks post-injury. Saline-treated wounds (Figure 3E) displayed typical fibrosis and hypertrophic scar tissue, indicated by the thickening of both the epidermis and dermis. The dermis contained dense connective tissue (deep blue color of the collagen by trichrome staining), multiple fibroblasts, no skin appendages such as hair shafts, sebaceous glands, smooth muscle cells, or adipocytes. In contrast, wounds that were treated with α-gal nanoparticles (Figure 3F) displayed restoration of the normal skin structure, including a normal thin epidermis and dermis consisting of loose connective tissue and partial reappearance of hair shafts, smooth muscle cells, and adipocytes [34]. This regeneration of normal skin structure suggests that the rapid complement-mediated recruitment of macrophages and the polarization of the recruited cells into pro-regenerative macrophages starts the regenerative processes before the default fibrosis and scar formation processes are initiated, thereby avoiding scar formation. Skin burns caused by thermal injury [33] or radiation [72] were also found to display an accelerated healing process in mice treated with α-gal nanoparticles, in comparison to saline-treated injuries. In addition, chronic wounds in anti-Gal-producing diabetic GT-KO mice healed within 12 days following α-gal nanoparticle treatment whereas saline treatment had no healing effects on such chronic wounds, characteristic of diabetes [49,67].

The mechanism that induces macrophages to mediate the scar-free regeneration of wounds in adult mice treated with α-gal nanoparticles rather than physiologic fibrosis, requires further elucidation. The studies above suggest that a combination of recruitment of monocyte-derived macrophages to wounds by C5a and C3a and anti-Gal-mediated Fc/Fcγ receptor binding of α-gal nanoparticles to the recruited macrophages (Steps 2 and 3 in Figure 1B) induces polarization into M2 macrophages which have pro-regenerative activity, orchestrating tissue regeneration (Step 4), similar to that in urodeles, mouse fetuses, and neonates. Since the mechanism of wound healing is common both to external and internal injuries, it is possible that the administration of α-gal nanoparticles into internal injuries or to surgical incision sites may result in the acceleration of the healing process and prevention of scar formation, similar to that observed in skin injuries.

## 5. α-Gal Nanoparticles Induce the Regeneration of Post-MI Myocardium in Adult Mice

The post-myocardial infarction (post-MI) healing of injured myocardium in adult-mice displays characteristics similar to those observed in healing of wounds. Pro-inflammatory polarized macrophages reaching the injured myocardium debride the tissue of dead cells. Subsequently, pro-reparative macrophages mediate repair by fibrosis and scar formation in both skin and heart injuries [12,13,14,15,16,17,73,74]. In contrast, injured ventricular walls in urodeles and mouse neonates display complete regeneration following activation of the complement system and migration of pro-regenerative macrophages into the injury site [8,22,23,24,31]. Therefore, it was of interest to determine whether post-MI injection of α-gal nanoparticles into the ischemic myocardium of anti-Gal-producing adult mice also induces regeneration of the injured ventricular wall and restores its normal function.

Post-MI myocardial treatment was evaluated in occlusion/reperfusion studies with anti-Gal-producing GT-KO mice [35]. After opening the chest and exposing the heart, the mid-left anterior descending (LAD) coronary artery was occluded by ligation for 30 min to simulate MI. Subsequently, the ligature was removed, allowing for reperfusion of the ischemic myocardium. The mice received two 10 µL injections of α-gal nanoparticles in saline (10 mg/mL) into the injured myocardium, within the territory supplied by reperfused LAD. In the post-MI control group, the mice received two 10 µL saline injections. The hearts were subjected to histological analysis at various time points.

Inspection of the hearts in the control mice after 28 days indicated that the MI procedure resulted in thinning of the ventricular wall and formation of transmural scarring made of fibrotic tissue in the left ventricular wall (Figure 4A). In contrast, after 28 days, post-MI ischemic myocardium injected with α-gal nanoparticles displayed near-complete regeneration of the left ventricular wall (Figure 4B) [35]. Only marginal fibrosis was detected in most α-gal nanoparticle-treated ventricular walls (Figure 4B). Regeneration of the injured myocardium also resulted in restoration of the contractile function of the left ventricle. Echocardiography analysis of pre-ligation left ventricular function displayed ~50% fractional shortening and a decrease to ~35% fractional shortening in both groups, 7 days post-MI. However, in the control group, the fractional shortening was ~35%, also 28 days post-MI, whereas in the α-gal nanoparticle-treated mice, the fractional shortening increased back to ~50% [35].

Monitoring the cellular changes in treated post-MI hearts at various time points demonstrated peak infiltration of pro-reparative macrophages into saline-treated control hearts on Day 4 post-MI, whereas on Day 7, the number of infiltrating macrophages was much lower (Figure 4A). Day 14 saline-treated hearts displayed significant thinning of the ventricular wall, disappearance of the infiltrating macrophages, and distinct initiation of fibrosis, indicated by the blue/gray trichrome staining of the tissue within the ventricular wall (Figure 4A) [35]. In the α-gal nanoparticle-treated hearts in Figure 4B, initial infiltration of pro-regenerative macrophages was observed on Day 4 at the two injection sites within the injured myocardium (marked by arrows). Peak infiltration of macrophages was observed on Day 7. A higher magnification of the Day 7 infiltrates indicated the presence of uninjured cardiomyocytes (large pink stained cells at the right part of Figure 4D), which are similar to those observed in an uninjured myocardium (Figure 4C). The multiple infiltrating macrophages that were recruited by α-gal nanoparticles are the purple-stained small cells in the upper left part of Figure 4D. In addition, a third group of large cells with basophilic cytoplasm, characteristic of proliferating cells, are observed in the lower left area of Figure 4D. The arrow identifies one of these large proliferating cells with a mitotic figure. The identity of the proliferating cells as “resident stem cells” in the heart [75], or as recruited mesenchymal stem cells, is not clear at present. Cell fate lineage studies of these cells may help in identifying the origins of the proliferating large cells in Figure 4D. Both infiltrating macrophages and the large proliferating cells disappeared by Day 14 and the ventricular wall displayed a near-complete regeneration of the injured myocardium. Only a residual, blue-stained, fibrotic tissue was observed in the left lower part of the ventricular wall (Figure 4B) [35]. This observation suggests that the proliferating cells presented in Figure 4D completed their differentiation into mature cardiomyocytes by the end of the second week post-MI. Similar near-complete regeneration of injured myocardium on Day 28 was observed in a total of 20 post-MI mice treated with α-gal nanoparticles [35].

The studies on the regeneration of ischemic myocardium in adult mice following occlusion/reperfusion of the LAD [35] suggest that α-gal nanoparticle-mediated regenerative processes may be similar to the scar-free regenerative processes in the wound healing of adult mice treated by these nanoparticles [33,34,49]. In both processes, the α-gal nanoparticles activate the complement system that initiates the recruitment of macrophages, which are further polarized into pro-regenerative macrophages that orchestrate the regeneration of the injured tissues, rather than fibrosis mediated by pro-reparative macrophages. The molecular and cellular differences between pro-reparative and pro-regenerative macrophages await further elucidation. Overall, the studies on α-gal nanoparticle-mediated regenerative processes in the injured skin and heart of adult mice mimic the physiologic regenerative processes observed in mouse neonates [22,23,24] and urodeles [1,8,10,29,30,31].

## 6. Nerve Regeneration in Injured Spinal Cord by α-Gal Nanoparticles

The significance of macrophages in regenerative processes in urodeles can also be observed in nerve regeneration. Studies on axonal regeneration of injured spinal cord in the axolotl indicated that this process is initiated following rapid macrophage migration into the lesion site [76]. The macrophages are large, filled with vacuoles containing lipid droplets [77], and have a morphology similar to that in Figure 2D [34,57]. The subsequent axonal regeneration proceeds at a rate of 0.05 mm per day [78]. This participation of macrophages in the early stages of spinal cord injury regeneration in the axolotl and the ability of α-gal nanoparticles to recruit macrophages to mouse nerves, as demonstrated in Figure 2C [57], raised the question of whether the recruitment of macrophages into spinal cord and peripheral nerve injuries by these nanoparticles could contribute to regeneration of such injuries in adult mice [28].

Regeneration of nerves requires regrowth of multiple sprouts from injured axons. These sprouts “attempt” to grow across the lesion and reconnect with distal targets. Inhibition of axonal sprouting and outgrowth, prevented by pathophysiologic processes such as fibrosis and scarring, irreversibly obstructs bridging of the sprouts over the lesion, ultimately resulting in loss of neurologic function [78,79,80]. Studies of the factors that contribute to axonal sprouting have indicated that this sprouting process and growth is coincidental and proportional to the extent of neovascularization within the lesion [80,81,82,83,84,85,86]. Therefore, treatments inducing local angiogenesis such as administration of VEGF have been posited to elevate the number of axonal sprouts growing along blood vessels within the lesion, thereby increasing the probability that some of the sprouts will “succeed” in connecting with distal targets [84,85,86]. In our studies, macrophages activated by anti-Gal-coated α-gal nanoparticles were found to secrete VEGF, as shown in Figure 1F [34]. These activated macrophages were further found to be in an M2 polarization state, as shown in Figure 2F [57,67,68]. Moreover, as shown in Figure 5G,H, macrophages infiltrating into crushed spinal cord injected with α-gal nanoparticles display M2 polarization as well [66]. We also found that microglia, the resident macrophages of the CNS, are similarly activated by α-gal nanoparticles and show many of the similar pro-regenerative characteristics of their blood borne counterparts [50]. Therefore, we hypothesized that administration of α-gal nanoparticles into crushed spinal cord in anti-Gal-producing GT-KO mice may result in regeneration of the spinal cord, due to localized VEGF secretion by the recruited and activated macrophages and microglia, similar to the steps presented in Figure 1B [66].

To determine the ability of α-gal nanoparticles to recruit pro-regenerative immune cells into the spinal cord, mice received either 0.5 μL intraparenchymal injections of saline or 0.5 μL injections of α-gal nanoparticles (10 mg/mL in saline). Histologic inspection of macrophages in the injected areas after 7 days was performed by immunostaining with an F4/80 antibody which specifically stains both macrophages and resident microglia. This staining demonstrated a much higher number of F4/80^+^ cells in spinal cords injected with α-gal nanoparticles than in those injected with saline (Figure 5A–F). These findings imply that recruitment of both macrophages and microglia by α-gal nanoparticles can occur within the spinal cord milieu, consistent with observations in other injection sites (Figure 2 and Figure 4B,D).

The hypothesis of α-gal nanoparticles promoting regeneration after spinal cord injury was studied in anti-Gal-producing GT-KO mice undergoing laminectomy followed by T9–T10 crush of the spinal cord. This injury modality is often encountered in human cases of spinal cord injury [87,88]. The polarization state and other characteristics of macrophage populations recruited into the injured spinal cord were studied by immunohistochemical analysis [66]. Previous studies [89,90,91,92,93,94,95] indicated that in spinal cord injuries, M1 macrophages facilitate phagocytosis and secrete pro-inflammatory cytokines such as interleukin-6 (IL-6) and tumor necrosis factor alpha (TNF-α). These macrophages perform cell debris clearance, and their persistence can also lead to cell death and axonal dieback through increased reactive oxygen species (ROS) generation. In contrast, M2 macrophages produce growth factors such as TGF-β and IL-10 and are implicated in tissue remodeling, spinal repair, and neuronal sprouting. Without timely replacement of the M1 to M2 macrophage phenotype in post-spinal cord injury (SCI), the healing process results in default fibrosis and scar formation, which leads to permanent prevention of axonal regeneration.

Arginase-1 expression (a marker of M2 macrophages), within and bordering the injury site, increased in the α-gal nanoparticle-treated group when compared to saline-injected mice at 7, 14, and 21 days post-injury (dpi) (Figure 5H). By 28 and 45 dpi, arginase-1 expression significantly decreased and was comparable in both groups. Analysis of a second marker of M2 macrophages, CD206, indicated that its expression in α-gal nanoparticle-treated mice also increased at 14, 21, and 28 dpi in comparison to saline-injected mice (Figure 5H) and later decreased by 28 and 45 dpi. In contrast, expression of the CD16/32 marker (characteristic of M1 macrophages) was higher in the saline-treated group than in the α-gal nanoparticle-treated group at all time points [66]. These findings, which are in agreement with the recruited macrophage M2 phenotype presented in Figure 2E,F, strongly suggest that macrophages recruited by α-gal nanoparticles within spinal cord injury sites display M2 phenotype markers, similar to those observed in subcutaneously implanted sponge discs containing these nanoparticles.

Previous studies have indicated that M2 macrophages secrete a variety of neurotrophins such as neurotrophin-3 (NT-3), brain-derived neurotrophic factor (BDNF), nerve growth factor (NGF), ciliary neurotrophic factor (CNTF), and leukemia inhibitory factor (LIF) which promote the survival and growth of neurons [18,96,97,98,99]. These studies led to the suggestion that modulation of the polarization state of macrophages in the SCI lesion may be used to promote nerve repair. This suggestion and the observation of extensive infiltration of M2 macrophages into spinal cord lesions treated with α-gal nanoparticles support the assumption that injection of these nanoparticles into nerve lesions may increase the probability of axonal regeneration instead of fibrosis and scarring [28].

Further immunohistological analysis of secondary injury markers also suggested that α-gal nanoparticles exerted some neuroprotection in the immediate days following spinal cord injury. For example, there was a noticeable decrease in the production of iNOS and BAX (pro-apoptotic) proteins with a concomitant increase in BCL-2 (anti-apoptotic) protein. These results were accompanied by a decreased presence of CD16/CD32 macrophages and an increased number of cells presenting CD31 (endothelial marker) and of cells producing VEGF at 28 and 45 dpi [66].

Visualization of axonal ingrowth into the lesion was quantified using neurofilament staining methods [66]. Increased neurofilament staining at 28 and 45 dpi within the lesions of α-gal nanoparticle-treated mice was found to be much higher than that in saline-treated mice, as the saline sections displayed very sparse neurofilamental staining within the lesions (Figure 6A). In the α-gal nanoparticle group, some axons can be observed completely traversing the lesion. Quantification of the neurofilaments indicated that the average axonal ingrowth into the lesion was significantly higher both at 28 and 45 dpi in α-gal nanoparticle-injected mice compared to the saline control (Figure 6B). Parallel immunohistochemistry of glial fibrillary acidic protein (GFAP) indicated a much lower number of astrocytes in the α-gal nanoparticle-treated lesions than in saline-treated lesions (Figure 6C). This indicates that the astrocyte populations in the saline group penetrated deeper into lesions, where they contributed to inflammatory processes leading to fibrosis. In contrast, fewer astrocytes migrated into the nanoparticle-treated lesion, suggesting lower fibrotic activity [66].

Collectively, the animal studies suggest that α-gal nanoparticles applied to spinal cord injuries in anti-Gal-producing mice promote axonal regeneration by ingrowth into the lesion. The proposed mechanism is similar to other tissue systems, in which the α-gal/anti-Gal complex activates the complement system, and via C5a and C3a cleavage peptides, induces migration of monocytes derived macrophages and microglia to the injection site (Figure 7). Binding of anti-Gal-coated α-gal nanoparticles to the recruited macrophages further induces polarization of these cells into M2 pro-regenerative macrophages that secrete VEGF—a growth factor that mediates extensive angiogenesis within the treated lesions. These observations are in accord with previous studies reporting that administration of VEGF to neural lesions of transected nerves induces localized angiogenesis and nerve regeneration after axotomy [79,80,82,83,84]. The newly formed multiple blood vessels facilitate the growth of many axonal sprouts which increase the probability of axonal bridging across the lesion and ultimately, improved functional outcomes. The unique therapeutic advantage of α-gal nanoparticle treatment of nerve regeneration is that the VEGF-producing pro-regenerative M2 macrophages that are recruited by these nanoparticles reside in the lesion for at least 3 weeks. Thus, this treatment enables the localized production of VEGF for several weeks, thereby ensuring prolonged enhancing effects on axonal ingrowth into the lesion. In view of the multiple neurotrophins produced by M2 macrophages [18,94,95,96,97,98,99], it will be of interest to determine whether the M2 macrophages that mediate axonal regeneration in SCI lesions treated with α-gal nanoparticles also produce these neurotrophins at the injection sites.

## 7. Discussion

In situ localized interaction between α-gal nanoparticles and the natural anti-Gal antibody offers a novel method for inducing regeneration of external and internal injuries. The studies in anti-Gal-producing mice demonstrate the recruitment and activation of macrophages which polarize into pro-regenerative M2 macrophages. These macrophages orchestrate the restoration of the pre-injury structure and function of injured skin [33,34,67,70,72], heart [35], and spinal cord injuries [66], and they prevent healing by the default mechanism of fibrosis and scar formation (Table 1). Wounds and burns treated with α-gal nanoparticles further displayed accelerated healing, resulting in restoration of the normal skin structure within 6–7 days, whereas healing of control wounds and burns took double that time [33,34]. A similar accelerated healing by regeneration was also observed in anti-Gal-producing pigs [71]. To our knowledge, the examples of the regenerative effects of α-gal nanoparticles in skin, heart, and spinal cord injuries are the only ones studied so far. It is reasonable that regeneration and accelerated healing of skin wounds and burns should be the initial treatments for establishment of efficacy and safety in clinical trials. Regeneration of injuries in the myocardium and spinal cord requires additional research for the optimization of these parameters, in anti-Gal-producing GT-KO mice and in large experimental animal models producing this antibody.

Treatment of wounds in humans with α-gal nanoparticles is likely to be safe. The phospholipid and cholesterol components of these nanoparticles are the same as those in humans. Only the α-gal glycolipid is not present in humans and is extracted from rabbit RBC membranes. Phase 1 clinical trials that included intra-tumoral injections of α-gal glycolipids extracted from rabbit RBC membranes, were found to be safe in treated patients with no indications of toxicity [100,101]. In addition, α-gal nanoparticles are highly stable, as indicated by their ability to bind the anti-Gal antibody. Such binding does not decrease even after 4 years of storage as a suspension at 4 °C or frozen at −20 °C. Moreover, the activity of α-gal nanoparticles dried on wound dressings and stored at room temperature does not diminish for at least 12 months. Treatment of wounds or burns with these nanoparticles may be performed by their application as a suspension, an aerosol, or as dried nanoparticles on wound dressings. In addition, these nanoparticles may be applied to injury sites in a semisolid medium such as a water-based ointment, hydrogel, plasma clot, or incorporated into biodegradable scaffold materials, such as collagen sheets [57]. The effective and safe treatment of skin injuries, such as that observed in anti-Gal-producing GT-KO mice, may suggest that a similar treatment might be considered also for accelerating the healing of internal and skin surgical incisions while minimizing or preventing fibrosis and scar formation.

One aspect of treatment with α-gal nanoparticles that should be evaluated in humans is its effects in individuals with α-gal syndrome. These individuals produce anti-Gal IgE following bites by the *Amblyomma americanum* tick (lone star tick) in the USA [102], or by other ticks in various continents [103,104,105], and suffer from allergic reactions to meat (beef, pork, and lamb). These allergic responses are caused by the interaction of the anti-Gal IgE with α-gal epitopes on glycolipids and glycoproteins released from the digested meat [102]. Skin tests with α-gal nanoparticles as allergens and dose escalation studies with these nanoparticles in the wounds of such patients may indicate whether this wound treatment induces allergic reactions in individuals with α-gal syndrome and requires additional anti-allergic treatment.

In addition to the potential clinical significance of α-gal nanoparticles, the research on regenerative induction in adult mice raises a significant basic biological question: In view of the ability of macrophages in mouse neonates to induce the regeneration of injured tissues such as the skin and myocardium, what is the cause of the loss of this ability in adult mice and its replacement with fibrosis and scar formation? The observed regeneration of the injured heart in urodeles [8] and neonates of mice [22,23,24] and pigs [25,26] is associated with macrophage activity and innate activation of the complement system [31]. Also, post-MI healing of the injured heart in adult mammals involves macrophage activity, but it results in fibrosis and scar formation [14,15,16,17]. All these observations suggest that macrophage-induced regenerative activity is associated with innate activation of the complement system and is an ancient evolutionary mechanism that has been conserved in amphibians and in mammalian neonates, but is suppressed in mammals within a few days after birth. The ability of anti-Gal/α-gal nanoparticles to interact in adult mice, activate the complement system, recruit macrophages, and polarize them into pro-regenerative macrophages, as demonstrated in this review, suggests that the ultimate effect of this interaction results in reactivation of suppressed genes in macrophages. The products of thes reactivated genes may orchestrate the healing process by the regeneration of injured tissue treated with α-gal nanoparticles.

A recent study [106] analyzed activated genes during regeneration of injured spinal cord in axolotl in comparison to activated genes in spinal cord injuries of adult mice that led to fibrosis and scar formation. This study demonstrated much longer activity of some immune-associated genes in the axolotl than similar genes in mice [106]. Future comparisons between activated genes in macrophages orchestrating regeneration in injured organs in axolotl and in adult mice treated with α-gal nanoparticles vs. control mice, may enable identification of the regeneration-associated genes that are active in urodeles, suppressed in adult mammals, and reactivated following α-gal nanoparticle treatment. Such studies will contribute to the development of methods for the replacement of default fibrosis and scar formation processes in injuries, with regenerative processes in tissues and organs.

## 8. Conclusions

One of the major immune-associated differences between the post-injury regenerative processes in urodeles and mouse neonates and the post-injury healing by fibrosis and scar formation in adult mice is the activation of the complement system in the two former groups, but not in the latter group. This led to the hypothesis that extensive localized complement activation in injury sites in adult mice may induce regeneration instead of the default fibrosis processes. Such complement activation is feasible by intra-injury application of α-gal nanoparticles which bind the anti-Gal antibody (a natural antibody constituting ~1% of immunoglobulins in humans). The C5a and C3a chemotactic peptides produced by anti-Gal/α-gal nanoparticle interaction direct the extensive recruitment of macrophages to the injury sites. The Fc/Fcγ receptor-mediated binding of anti-Gal-coated α-gal nanoparticles to the recruited macrophages activates these cells to polarize into pro-regenerative M2 macrophages, secrete pro-regenerative cytokines/growth factors, and recruit stem cells. Studies in anti-Gal-producing adult mice indicated that these activated macrophages mediate accelerated scar-free regeneration of wounds and of post-MI injured myocardium, similar to the regeneration observed in urodeles. In addition, α-gal nanoparticles injected into spinal cord injuries induce extensive axonal ingrowth into the lesion. Since the natural anti-Gal antibody is produced in large amounts in humans, these regenerative α-gal nanoparticle treatments may be considered for further evaluation as future therapies of external and internal injuries in a variety of clinical settings.

## Figures and Tables

**Figure 3 nanomaterials-14-00730-f003:**
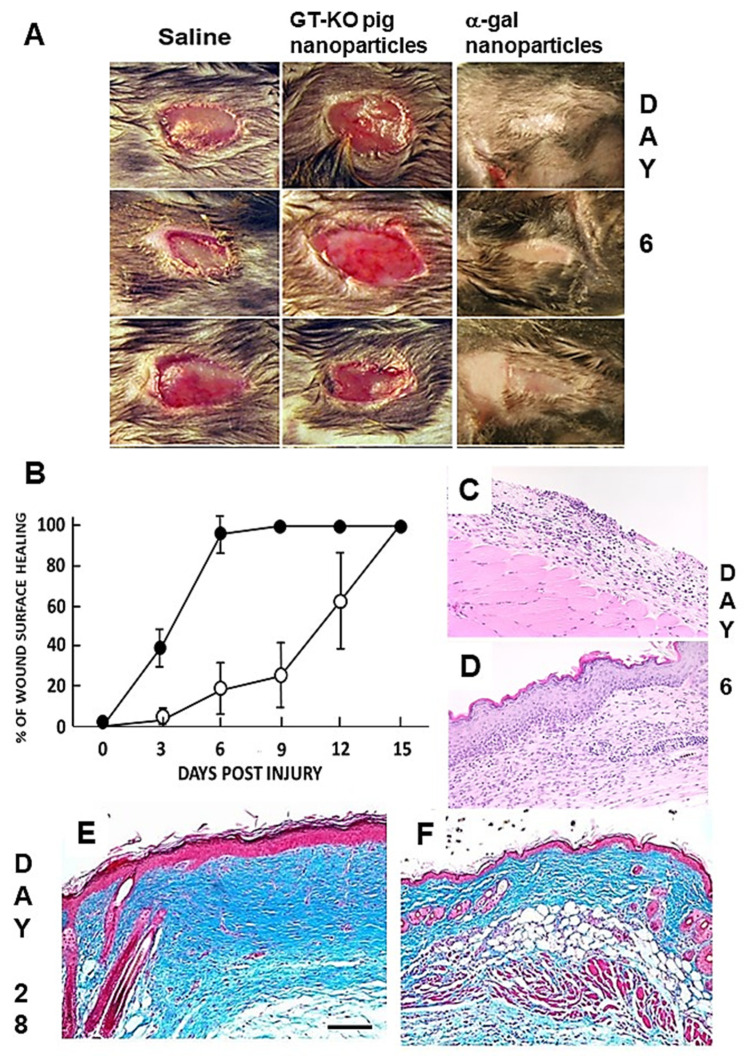
Accelerated scar-free wound regeneration in anti-Gal-producing GT-KO mice by 10 mg α-gal nanoparticles. (**A**) Gross morphology of 3 representative excisional full-thickness oval skin wounds in 3 groups (9 × 6 mm), following treatment with saline, nanoparticles lacking α-gal epitopes, or α-gal nanoparticles, viewed on Day 6 post-injury. Whereas no significant healing was observed in mice treated with saline or with nanoparticles lacking α-gal epitopes, α-gal nanoparticle treatment induced the complete covering of wounds with regenerating epidermis. (**B**) The healing of wounds at various days post-wounding, as measured by the percentage of wound area covered by regenerating epidermis following treatment with saline (○) or with α-gal nanoparticles (●). Mean ± SD of n = 5–8 mice on all days except n = 20 mice on Day 6. (**C**) Histology of a representative saline-treated wound on Day 6 displaying a complete lack of regenerating epidermis (H&E ×100). (**D**) Histology of a representative α-gal nanoparticle-treated wound on Day 6 displaying complete regeneration of the epidermis (H&E ×100). (**E**) A representative saline-treated wound on Day 28, displaying fibrosis and scar formation. (**F**) A representative α-gal nanoparticle-treated wound on Day 28 displaying regeneration of the normal structure of the skin (trichrome ×100 staining collagen blue in (**E**,**F**)). The bar in (**E**) represents a scale of 100 μm and is common also to Figures (**C**,**D**,**F**). Adapted with permission from Ref. [57]. 2018, Elsevier.

**Figure 4 nanomaterials-14-00730-f004:**
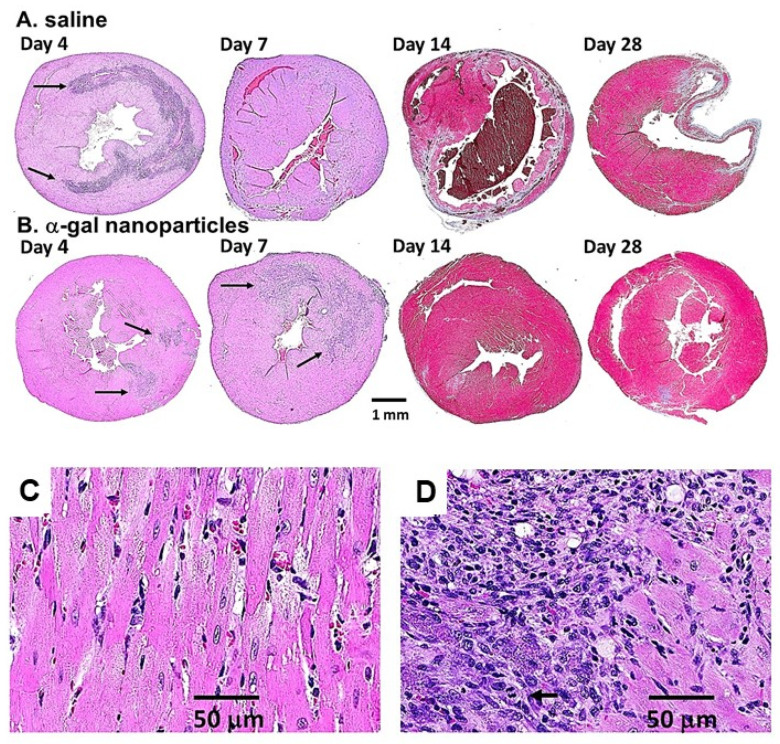
Post-MI regeneration of injured myocardium by α-gal nanoparticles at various time points. (**A**) Fibrosis and scar formation in hearts injected with saline post-MI. (**B**) Regeneration of injured myocardium in α-gal nanoparticle-treated hearts. Sections of Day 4 and 7 hearts were stained with H&E (arrows indicate macrophage infiltration), and of Day 14 and 28, with trichrome (fibrotic tissues stained blue/gray, uninjured myocardium red, and RBCs brown). Note the appearance of fibrosis on Day 14 and clear scar formation on Day 28 in saline-treated hearts, whereas in α-gal nanoparticle-treated hearts, near-complete regeneration of injured myocardium is observed on Day 14, as well as on Day 28. Representative hearts from n = 2 on Days 4, 7, and 14, n = 10 in saline-treated control mice, and n = 20 of α-gal nanoparticle-treated mice on Day 28. (**C**) Histology of healthy myocardium. (**D**) Cell types appearing on Day 7 post-MI in mouse myocardium undergoing regenerative processes following treatment with α-gal nanoparticles. Healthy cardiomyocytes in the right half of the figure, infiltrating macrophages in the upper left quarter, and large cells with basophilic cytoplasm, suggesting proliferation, in the lower left quarter of the figure. The arrow marks a cell in mitosis (H&E, ×400). Adapted with permission from Ref. [35]. 2021, Frontieres.

**Figure 5 nanomaterials-14-00730-f005:**
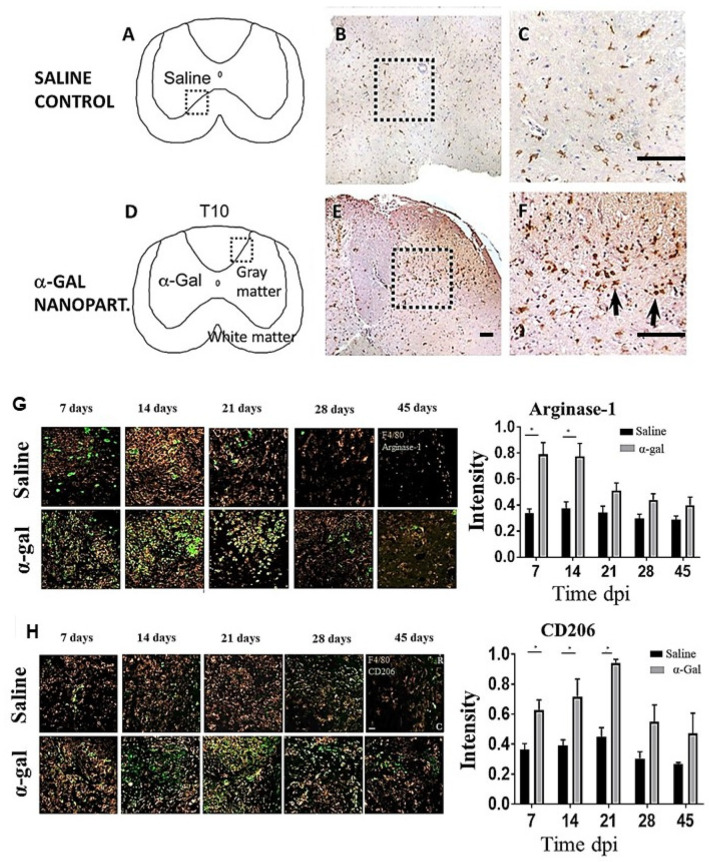
Staining of microglia/macrophages in intact spinal cord of GT-KO mice injected with either saline or α-gal nanoparticles. Saline-injected spinal cord (**A**–**C**). α-Gal nanoparticle-injected spinal cord (**D**–**F**). The dotted rectangles represent the area of the micro-injection. Black arrows in (**F**) highlight marked amoeboid morphology of activated F4/80 cells at the injection site in the α-gal nanoparticle group. (**G**) Expression of arginase 1 (green) and F4/80 (brown) in macrophages in crushed spinal cord at various days post-injury (dpi) and injected with saline or α-gal nanoparticles. (**H**) Expression of CD206 (green) and F4/80 (brown) in macrophages in crushed spinal cord at various dpi and injected with saline or α-gal nanoparticles. Co-localization between these markers and F4/80 were used to ascertain the macrophage/microglial phenotype (H&E and F4/80 staining of macrophages brown ×4 (**B**,**E**) and ×20 (**C**,**F**)). Scale bars = 100 μm. Quantification of the normalized fluorescence intensity of the respective markers are shown in the right-hand panels, n = 4 animals per group. * *p* < 0.05. Adapted with permission from Ref. [66]. 2024, Springer.

**Figure 6 nanomaterials-14-00730-f006:**
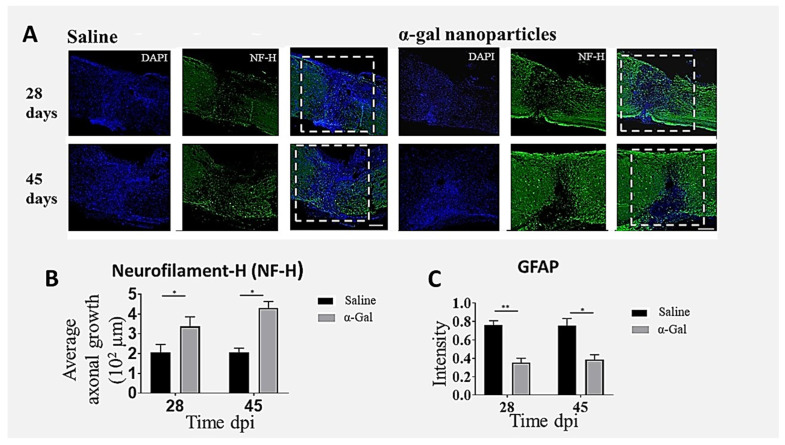
Axonal ingrowth into spinal cord lesions treated with α-gal nanoparticles and expression of GFAP in these lesions. (**A**) Effects of saline or α-gal nanoparticle injection into crushed spinal cord lesions on axonal growth into the lesions, evaluated in longitudinal spinal cord sections of mice at 28 or 45 dpi, following staining with the neurofilament specific antibody (NF-H) and with DAPI, which stains cell nuclei. The dashed rectangles represent the injury site. Scale bars = 100 μm. (**B**) Average axonal growth within the lesion at 28 and 45 dpi obtained from NF-H-stained images. (**C**) Mean fluorescence intensity of GFAP found in the astroglial cytoskeleton and stained antibody to GFAP. The extent of staining was normalized against uninjured regions. n = 3–4 animals per group. * *p* < 0.05, ** *p* < 0.005. Adapted with permission from Ref. [66]. 2024, Springer. The improvements in histological outcomes were further in concordance with the return of some gross and fine sensorimotor functions after spinal cord crush. The α-gal nanoparticle-treated mice demonstrated higher locomotor and exploratory activity via an open field assay, fewer foot slips on a tapered balance beam apparatus, and improved sensory withdrawal reflex (no hyperalgesia) measured by the electric von Frey method [66].

**Figure 7 nanomaterials-14-00730-f007:**
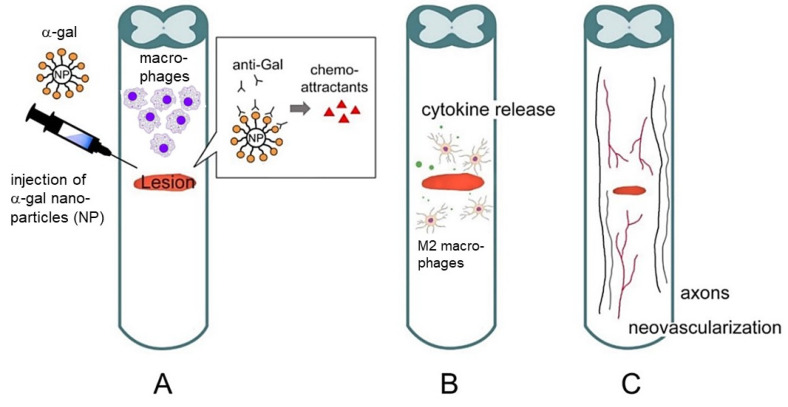
Suggested therapy for the regeneration of injured nerves by α-gal nanoparticles. (**A**) Anti-Gal released from ruptured blood vessels binds to α-gal nanoparticles injected into the injured spinal cord. This anti-Gal/α-gal nanoparticle interaction activates the complement system, resulting in the production of complement cleavage chemotactic peptides such as C5a and C3a which recruit macrophages and microglia. These nanoparticles further polarize the recruited macrophages into M2 cells. (**B**) Recruited M2 macrophages and microglia secrete pro-regenerative cytokines/growth factors, including VEGF. (**C**) The released VEGF and other cytokines/growth factors induce neovascularization and extensive axonal sprouting and a microenvironment conducive for regeneration. The multiple axonal sprouts bridge over the lesions and increase the probability of reconnection between the proximal and distal sections of the severed axons, thereby mediating the regeneration of neural function. Adapted with permission from Ref. [66]. 2024, Springer. It is of note that since the model above is of a crushed spinal cord, rather than of clear transected nerves, we cannot determine at present the relative proportion of axonal ingrowth due to regeneration vs. axonal sparing. It could be that both mechanisms are mediated by α-gal nanoparticle treatment and contribute to the observed axonal ingrowth.

**Table 1 nanomaterials-14-00730-t001:** Summary of studies on the induction of accelerated healing and regeneration of injuries in various tissues by treatment with α-gal nanoparticles.

Tissue Injury	Experimental Animal	Treatment Results	References
1. Skin Wound	GT-KO mouse ^a^	Accelerated healing by scar-free regeneration	[34,70]
2. Chronic Skin Wound (diabetes)	GT-KO mouse	Chronic wound healing	[49,67]
3. Skin Burn and Radiation	GT-KO mouse	Accelerated healing	[33,68,72]
4. Skin Wound	GT-KO pig ^b^	Accelerated healing by scar-free regeneration	[71]
5. Myocardial Ischemia (post-MI)	GT-KO mouse	Near-complete myocardial regeneration	[35]
6. Spinal Cord Crush	GT-KO mouse	Accelerated axonal growth and scar-free regeneration	[66]

^a^ α1,3galacrosyltransferase knockout mouse. ^b^ α1,3galacrosyltransferase knockout pig.

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
