# Peer review of "Regeneration in Mice of Injured Skin, Heart, and Spinal Cord by α-Gal Nanoparticles Recapitulates Regeneration in Amphibians"

_nanomaterials, 2024, doi:10.3390/nano14080730_

Round 1
Reviewer 1 Report
Comments and Suggestions for Authors
Uri Galili et al. submitted an interesting review upon α-Gal nanoparticles for tissue regeneration application. The topic was of significance nowadays, and therefore the review might arouse a certain impact in its field. However, there were some issues pending addressed, and a Major Revision must be conducted. The detailed comments are as follows.
1) The Abstract was too long, which contained about 330 words. Please substantially reduce it to about 200 words. Only most eye-catching information was needed for an abstract.
2) Please supplement Keywords below the Abstract.
3) Some writing styles in the Introduction could be optimized. The authors used several “one of the most…” expressions at the beginning of paragraphs. This largely implied that there were many other items you omitted. Please consider to change the expressions.
4) Section 2 must be revised. Firstly, it was unclear what the “hypothesis” was. Could you please define it, and better put it as the subtitle for this section? Secondly, the length of this section was rather shorter than others, making it awkward. Thirdly, the process was a bit difficult to interpret. A scheme was suggested to illustrate the hypothesis.
5) In Section 4~6, the authors summarized the application of α-Gal nanoparticles. The reviewer wondered that if there were other applications? Like muscle regeneration? If so, please consider to at least make a brief summary of the other regeneration applications.
6) Before the Conclusion Section, it was advisable to add a Discussion Section to convey the authors own opinion on the future directions of α-Gal nanoparticles.
7) A few underlines appeared in the Reference lists. Perhaps remove them.
Author Response
We thank the reviewer for the comments which improve the manuscript. Enclosed is the manuscript revised according to the suggested comments. Our response to the comments is beneath each comment and includes information on the line(s) containing the corresponding revision. The revisions were highlighted by the “Track changes” program in WORD and and a "clean" manuscript may be generated with the “Accept all changes” command.
1) The Abstract was too long, which contained about 330 words. Please substantially reduce it to about 200 words. Only most eye-catching information was needed for an abstract.
RESPONSE: As suggested, the size of the Abstract was decreased to 200 words.
2) Please supplement Keywords below the Abstract.
RESPONSE: As suggested, the Keywords were added below the Abstract.
3) Some writing styles in the Introduction could be optimized. The authors used several “one of the most…” expressions at the beginning of paragraphs. This largely implied that there were many other items you omitted. Please consider to change the expressions.
RESPONSE: As suggested, the words “one of the most…” were changed in lines 96 and 104, to fit the context.
4) Section 2 must be revised. Firstly, it was unclear what the “hypothesis” was. Could you please define it, and better put it as the subtitle for this section? Secondly, the length of this section was rather shorter than others, making it awkward. Thirdly, the process was a bit difficult to interpret. A scheme was suggested to illustrate the hypothesis.
RESPONSE: As suggested, the “Hypothesis” section was clarified by increasing the details explaining the hypothesized process. In addition, the Steps 1-4 in the Hypothesis are described to clearly explain the corresponding parts of the illustration (included as Figure 1B). The revised longer text of the Hypothesis is included in lines 168-205. In addition, the Figure presenting the hypothesis (Figures 1A and 1B) was enlarged and presented in higher definition in comparison to the previous submission, so that the small font within the rectangles, describing carbohydrate chains, is more legible. A scheme of the hypothesis was added in lines 197-205.
5) In Section 4~6, the authors summarized the application of α-Gal nanoparticles. The reviewer wondered that if there were other applications? Like muscle regeneration? If so, please consider to at least make a brief summary of the other regeneration applications.
RESPONSE: To our knowledge, the alpha-gal nanoparticles have been so far produced only by Dr. Uri Galili and colleagues and by no other research groups. The only injured tissues shown to regenerate in adult mice by alpha-gal nanoparticles treatment have been skin, heart and spinal cord, as described in this review and further indicated in line 613. The study of muscle regeneration, suggested by the reviewer, was performed several years ago by U. Galili and colleagues, causing ischemic damage in hind leg muscles of anti-Gal producing mice, by blocking blood flow. We found that the regeneration of the injured muscles in control mice (treated with saline) was not significantly longer than that observed in nanoparticles treated muscles (unpublished information). Thus, the healing of skeletal muscles differs from that of cardiomyocytes and is not suitable for demonstrating the regenerative effects of alpha-gal nanoparticles. It is of note that the accelerated healing of injured skin in healthy and diabetic GT-KO mice was independently demonstrated in the plastic surgery research lab of Dr. Jason Spector at Cornell Medical School in New York, using alpha-gal nanoparticles provided by U. Galili. These studies are mentioned in lines 243, 328 and 344, and described in references, 67, 70 and 72.
6) Before the Conclusion Section, it was advisable to add a Discussion Section to convey the authors own opinion on the future directions of α-Gal nanoparticles.
RESPONSE: As suggested, a Discussion Section reflecting the views of the authors regarding future directions, was added in lines 611-683 and the corresponding added references #100-106.
7) A few underlines appeared in the Reference lists. Perhaps remove them.
RESPONSE: The former submitted manuscript and the present one have no underlines in the references section. If such underlines appear they may originate in the processing of the manuscript in the editorial office.
Reviewer 2 Report
Comments and Suggestions for Authors
The manuscripts reviewed the α-gal nanoparticle in the regeneration of injuries, especially skin, heart, and spinal cord of mice without scar formation as in amphibians. The authors have reviewed how biodegradable α-gal nanoparticles are involved in the process of healing to avoid fibrosis formation in mice. According to the review manuscript, the interaction between α-gal nanoparticles and natural anti-gal antibodies can activate the complement system to accelerate the healing process and scar-free formation. The authors have reviewed several sub-topics to elucidate the possible mechanism in scar-free formation and their experimental demonstrations following the steps of the hypothesized mechanism to accelerate the healing process while obtaining scar-free formation or avoiding default fibrosis formation through localized complement activation. In the subsequent sub-topics, the authors focused on different types of injuries that can be healed through the activation of the localized complement system with anti-gal antibody coated α-gal nanoparticles by inducing rapid recruitment of macrophages that are polarized into M2 pro-regenerative macrophages to accelerate wound healing and scar-free formation. In this review manuscript, the regeneration of injured mice is not limited to skin, but myocardium and nerve regeneration by α-gal nanoparticles were also discussed in detail. The manuscript systematically and specifically reviews the α-gal nanoparticles to treat different injuries in mice to elucidate the hypothesis of working principles of the α-gal nanoparticles to accelerate the healing process and scar-free formation. The manuscript is ready to publish in its current form without any revision.
Author Response
We thank the reviewer for recommending the acceptance of the manuscript for publication in "Nanomaterials" without any revision.
Reviewer 3 Report
Comments and Suggestions for Authors
The reviewer received the text of a review-type manuscript entitled “Regeneration in mice of injured skin, heart, and spinal cord by alpha-gal nanoparticles recapitulates regeneration in amphibians,” prepared by a team of 3 authors.
The text of the manuscript reviews aspects of the mechanisms of healing damage of biological tissues from the point of view of the use of alpha-gal nanoparticles. A description of research aimed at restoring various tissues, including neuronal tissues, is provided, which will be of great interest to readers whose areas of scientific interest are close to the tasks of regenerative medicine.
In my personal opinion, both as a reader and as a reviewer, the article is for the most part presented in a finished form, close to publication. I can only recommend systematizing the data of the analyzed studies in the form of a table, which will clearly show which animal models were used, for what purpose and what the main result was. Such a table would be useful from the point of view of reader convenience and general perception of information.
Overall, I can recommend the article for acceptance with minor modifications.
Author Response
I can only recommend systematizing the data of the analyzed studies in the form of a table,
RESPONSE: The suggested table was added as Table 1 in page 20 and the reference to this table is in line 615, within the text of the Discussion requested by Reviewer #1.
Round 2
Reviewer 1 Report
Comments and Suggestions for Authors
Thanks for your revision.